# Differences in Optimal Platelet Reactivity after Potent P2Y12 Inhibitor Treatment in Acute Coronary Syndrome Patients Undergoing Percutaneous Coronary Intervention

**DOI:** 10.3390/jcm11092480

**Published:** 2022-04-28

**Authors:** Kai Song, Xuan Jin, Moo-Hyun Kim, Jia-Xin Li, Cai-De Jin, Song-Lin Yuan, Zhao-Yan Song, En-Ze Jin, Kwang-Min Lee, Kyung-Hee Lim, Young-Rak Cho

**Affiliations:** 1Department of Cardiology, Dong-A University Hospital, Busan 49201, Korea; songkaihmu@163.com (K.S.); jinxuan0819@daum.net (X.J.); ysl844959829@daum.net (S.-L.Y.); songzhaoyan168@gmail.com (Z.-Y.S.); tnt849@hanmail.net (K.-M.L.); jinjinsama@naver.com (K.-H.L.); nephrone@dau.ac.kr (Y.-R.C.); 2Department of Cardiology, Huizhou Third People’s Hospital, Guangzhou Medical University, Huizhou 516002, China; 3Department of Cardiology, Fourth Affiliated Hospital of Harbin Medical University, Harbin 150001, China; lijiaxinhmu@163.com (J.-X.L.); enzejin@163.com (E.-Z.J.); 4Department of Cardiology, Affiliated Hospital of Zunyi Medical University, Zunyi 563003, China; jincaide@zmu.edu.cn

**Keywords:** optimal platelet reactivity, P2Y12 inhibitors, VerifyNow, multiple electrode aggregometry, light transmittance aggregometry

## Abstract

Background: East Asian patients receiving treatment with the potent P_2_Y_12_ inhibitors prasugrel or ticagrelor experience more potent platelet inhibition than with clopidogrel. Methods: This study investigated differences in OPR rates with reduced doses of prasugrel (*n* = 38) or ticagrelor (*n* = 40) for maintenance therapy in 118 Korean ACS patients who had undergone PCI, in comparison to conventional-dose clopidogrel (*n* = 40). We assessed drug responses at one- and three-months post-PCI with VerifyNow and multiple electrode aggregometry assays. Results: At the one-month period, patients receiving standard-dose prasugrel or ticagrelor had lower platelet reactivity as determined by the three assays than those receiving the conventional dose of clopidogrel (VN: *p* = 0.000; MEA: *p* = 0.000; LTA: *p* = 0.000). At the 3-month point, platelet reactivity was lower in those receiving reduced-dose prasugrel or ticagrelor than the clopidogrel-treated patients (VN: *p* = 0.000; MEA: *p* = 0.012; LTA: *p* = 0.002). Prasugrel resulted in significantly lower platelet inhibition than ticagrelor as determined by VN and LTA (VN: *p* = 0.000; LTA: *p* = 0.003). At three months, there was a significant overall difference in OPR among the three groups when measured by VN (*p* < 0.001), but not when measured by MEA (*p* = 0.596). OPR in the reduced-dose prasugrel group was not significantly different to the clopidogrel group at three months (VN: *p* = 0.180; MEA: *p* = 0.711). OPR in the reduced-dose ticagrelor group was similar to clopidogrel as determined by MEA at three months, but was different when assessed by VN (VN: *p* = 0.000; MEA: *p* = 0.540). Compared to standard-dose, the reduced-dose prasugrel OPR rate was significantly increased (VN: *p* = 0.008; MEA: *p* = 0.020). Conclusions: OPR values for reduced-dose prasugrel and conventional-dose clopidogrel at three months were similar but higher than for reduced-dose ticagrelor as determined by VN, but no differences were noted by MEA. The MEA assay might have less sensitivity and consistency than the VN assay. Further studies are needed to explore this discrepancy.

## 1. Introduction

Dual antiplatelet treatment (DAPT) with aspirin and a P_2_Y_12_ receptor antagonist is the standard antithrombotic strategy in acute coronary syndrome (ACS) patients undergoing percutaneous coronary intervention (PCI), as well as those not undergoing PCI [1]. Prasugrel is an investigational thienopyridine prodrug that is metabolized into an active metabolite [1,2,3]. Ticagrelor is a reversible direct-acting oral antagonist of the adenosine diphosphate receptor P_2_Y_12_, providing more consistent P_2_Y_12_ inhibition [4]. Compared to clopidogrel, the newer P_2_Y_12_ inhibitors are associated with a significantly lower incidence of ischemic events without an increased risk of major bleeding [5]. In the TRITON-TIMI 38 and PLATO trials, prasugrel and ticagrelor significantly reduced ischemic events, particularly one month after PCI when compared to clopidogrel [6,7]. However, the rate of bleeding events in East Asian patients appears to be higher than in the western population for standard doses of the novel potent P_2_Y_12_ inhibitors [8]. With the East Asian Paradox hypothesis, we are trying to determine an optimal dose as with prasugrel [9]. Ticagrelor 45 mg bid might be a good option for patients with low body weight or elderly patients. It is therefore essential for physicians of patients of East Asian ethnicity to select the optimal treatment, which can include reduced doses of prasugrel/ticagrelor or de-escalation to clopidogrel. The present study sought to compare standard and reduced doses of prasugrel and ticagrelor with clopidogrel in the maintenance phase of PCI in Korean ACS patients. We hypothesized that ticagrelor would elicit greater platelet inhibition than prasugrel and clopidogrel.

## 2. Materials and Methods

### 2.1. Study Design and Population

This study was a platelet function sub-analysis of the HOPE–TAILOR trial (Half dose of Prasugrel and Ticagrelor in Platelet Response to Acute Coronary Syndrome; URL: Available online: https://www.clinicaltrials.gov (accessed on 24 October 2016); Unique identifier: NCT02944123) [10].

For inclusion criteria, we consecutively screened patients aged ≥18 years on DAPT with aspirin and a P2Y12 inhibitor, having received a loading dose (60 mg of prasugrel, 180 mg of ticagrelor or 300–600 mg of clopidogrel) during index admission or at the time of PCI. All patients were discharged with prescriptions for one-month standard-dose DAPT treatment: 100 mg once daily aspirin plus prasugrel 10 mg once daily, 100 mg once daily aspirin plus ticagrelor 90 mg twice daily or 100 mg once daily aspirin plus clopidogrel 75 mg once daily, followed by half-dose prasugrel 5 mg once daily or ticagrelor 45 mg twice daily for maintenance treatment, but with maintenance of the clopidogrel dose without reduction.

The exclusion criteria were as follows: (1) low body weight (<60 kg); (2) use of glycoprotein IIb/IIIa inhibitors tirofiban or eptifibatide within 24 h before or during PCI or abciximab within 10 days before or during PCI; (3) daily treatment with nonsteroidal anti-inflammatory drugs; (4) contraindication to aspirin or any of the study drugs; (5) current treatment with drugs interfering with CYP3A4 metabolism; (6) a history of stroke or transient ischemic attack; (7) gastrointestinal bleeding within the last 6 months, bleeding diathesis, platelet count <100,000/mm^3^; (8) known chronic renal insufficiency or hepatic dysfunction; or (9) known severe chronic obstructive pulmonary disease or bradycardia.

### 2.2. Blood Sampling

Venous blood samples for platelet function tests were obtained at two timepoints: 30 ± 7 days and 90 ± 7 days after PCI. After confirming that the patient was ingesting the study drugs, blood samples collected in the morning or afternoon and were processed by trained laboratory physicians who did not have access to the results. Platelet function assessments included VerifyNow (VN, Accumetrics, San Diego, CA, USA), multiple electrode aggregometry (MEA, Dynabyte Medical, Munich, Germany) and light transmittance aggregometry (LTA, Chrono-Log, Havertown, PA, USA).

### 2.3. Platelet Function Measurements

#### 2.3.1. VerifyNow

The platelet function test was performed on an outpatient basis at clinic times between 9:00–12:00 and 13:30–16:30 after confirming administration of the antiplatelet drug administration. The VN assay is a point-of-care (POC), turbidimetric assay that measures platelet function and was used according to manufacturer’s instructions [11]. Within the cartridge of the VerifyNow P2Y12 assay is a channel that measures inhibition of the adenosine diphosphate (ADP) P2Y12 receptor. This channel contains ADP as a platelet agonist and prostaglandin E1 as a suppressor of intracellular-free calcium levels to reduce the nonspecific contribution of ADP binding to P2Y12 receptors. The VN results are expressed in P2Y12 reaction units (PRU). The cutoff values used to define OPR were between 85 to 208 PRUs [12].

#### 2.3.2. Multiplate Analyzer

MEA is a semi-automated POC system, assessing platelet reactivity in whole blood, and was used according to the manufacturer’s instructions [13]. The output values are expressed in arbitrary aggregation units (AU). The cut-off values used to define OPR were between 19 to 46 AU [14]. MEA was performed with a Multiplate Analyzer (Dynabyte Medical, Munich, Germany). Specifically, the adhesion and aggregation of platelets on sensor surfaces enhances the electrical resistance between two sensor electrodes. We used an ADP test (6.4 µM ADP) to monitor the antiplatelet effects of DAPT, mostly targeting clopidogrel. In the test cuvette, whole blood (300 µL) was diluted (1:2 vol/vol) with 0.9% NaCl solution for 6.4 µM, and ADP was stirred in for 3 min at 37 °C, before ADP in the absence of PGE1 was added, and the increase in electrical impedance was recorded continuously for 6 min and converted into arbitrary aggregation units (AU). Approximately 8 AU corresponds to 1 Ohm. The means of the two independent determinations were expressed as the area under the curve of aggregation tracing (AUC) in AU· min. The manufacturer recommends the use of arbitrary units (U) to simplify the expression of results (1 U = 10 AU · min = 1 AUC).

#### 2.3.3. Light Transmittance Aggregometry

According to the LTA standard protocol, blood samples were drawn into a 3.2% sodium citrate-containing tube (Greiner Bio-One GmbH, Frickenhausen, Germany) and processed within 2 h [15]. Platelet-rich plasma was prepared by centrifugation at 120× *g* for 10 min. After the collection of platelet-rich plasma, platelet-poor plasma was obtained from the remaining specimen by recentrifugation at 1200× *g* for 10 min. The platelet-rich plasma was then adjusted to a platelet count of 250,000 per ml by adding platelet-poor plasma as needed. Light transmission was calibrated by a cuvette with platelet-rich plasma which was normalized as 0% and a second cuvette containing platelet-poor plasma that was normalized as 100%. Platelet function was measured after the addition of 10 mL adenosine diphosphate (ADP), before the curves were recorded for 6 min. The results are expressed as maximum platelet aggregation (MPA) within 6 min. At present, OPR measurements by LTA remain controversial. LTA was used to compare the other two assays.

### 2.4. The Definitions of OPR, HPR and LPR

High platelet reactivity (HPR) was defined as PRU > 208, while low platelet reactivity (LPR) was defined as PRU < 85 assessed by VN [16]. High platelet reactivity (HPR) was defined as AU > 46, while low platelet reactivity (LPR) was defined as AU < 19 assessed by MEA [12].

### 2.5. Statistical Analysis

For baseline characteristics, categorical variables are expressed as frequencies and percentages, and continuous variables as mean ± standard deviation. If application conditions were not fulfilled, the qualitative variables were compared using the chi-square test or Fisher’s exact test. The P2Y12 antagonist quantitative variables were compared with the ANOVA method between groups. The Pearson method was used to correlate antiplatelet effects measured by the two assays. To perform simple linear regression analysis, we first confirmed whether the data satisfied assumptions of linearity, independence, normality and homoscedasticity. For the clinical outcomes, we analyzed six-month major adverse cardiovascular events (MACE) and bleeding events using the Kaplan–Meier method, with a log-rank *p*-value. The hazard ratios (HRs) and 95% confidence intervals (CIs) were calculated with Cox proportional hazard methods comparing the three groups. Landmark analyses in the overall population and in prespecified subgroups were conducted, setting the landmark point at one month. *p* values < 0.05 were considered statistically significant for all comparisons using SPSS software version 22.0 (SPSS Inc., Chicago, IL, USA). A comparison of platelet reactive responses to prasugrel, ticagrelor and clopidogrel was undertaken with GraphPad Prism software (version 8.0.1; GraphPad, Inc., San Diego, CA, USA). Three groups of optimal platelet reactivity variation Sankey diagrams were generated using Origin software (version 2020; OriginLab, Inc., Northampton, UK).

## 3. Results

### 3.1. The Baseline Characteristics of Study Patients

From December 2016, through November 2018, 118 patients were enrolled in the analysis (losses were due to patients’ infeasibility to visit at the scheduled times or failure to obtain a suitable blood sample to measure platelet function), of which 40 (33.9%), 38 (32.2%) and 40 (33.9%) patients were treated with clopidogrel, prasugrel and ticagrelor, respectively. Table 1 summarizes the baseline demographics, and procedural and angiographic results for the population treated with the P2Y12 antagonist. Baseline results were slightly different between the groups. The clopidogrel-treated patients were slightly older on average than the other groups. More male patients were enrolled. Unstable angina was more frequent among the clopidogrel-treated patients, and ST-segment elevation myocardial infarction (STEMI) was more frequent in the ticagrelor-treated group. There was no significant difference in post PCI base values among the three groups.

### 3.2. Pharmacodynamic Effect of Standard- and Reduced-Dose New P2Y12 Inhibitors

At the one-month period, patients receiving standard-dose prasugrel or ticagrelor had lower platelet reactivity as determined by the three assays than those receiving the conventional dose of clopidogrel (VN: 41.8 ± 41.5 vs. 20.0 ± 25.3 vs. 161.8 ± 68.9, *p* = 0.000; MEA: 12.7 ± 5.9 vs. 15.6 ± 5.6 vs. 22.1 ± 11.8, *p* = 0.000; LTA: 6.2 ± 10.1 vs. 2.9 ± 4.9 vs. 13.4 ± 15.6, *p* = 0.000, Figure 1, Table 2). Compared to ticagrelor, there was significantly less platelet inhibition achieved with prasugrel determined by VN (*p* = 0.007), but the opposite result was obtained when determined by MEA (*p* = 0.031), and no change was shown with LTA (*p* = 0.080). At the 3-month point, platelet reactivity was lower in those receiving prasugrel or ticagrelor than the clopidogrel-treated patients (VN: 94.8 ± 57.0 vs. 31.0 ± 34.5 vs. 156.8 ± 66.1, *p* = 0.000; MEA: 16.8 ± 5.5 vs. 17.4 ± 4.7 vs. 21.9 ± 12.0, *p* = 0.012; LTA: 10.3 ± 9.4 vs. 4.1 ± 8.2 vs. 12.2 ± 13.4, *p* = 0.002). Prasugrel resulted in significantly less platelet inhibition than ticagrelor as determined by VN and LTA (VN: *p* = 0.000; LTA: *p* = 0.003). However, no difference in findings was determined when measured with MEA (*p* = 0.638). Prasugrel and clopidogrel had similar platelet inhibition at the three-month timepoint (*p* = 0.470).

At one month, OPR rates were markedly lower with both prasugrel and ticagrelor compared to clopidogrel as determined by VN (21.1%:0.0%:57.5%; *p* = 0.001); OPR rates for prasugrel and ticagrelor were prominently lower compared to clopidogrel determined by MEA (Table 3). After 3 months, OPR rates for prasugrel and clopidogrel were similar, and both were markedly higher compared to ticagrelor determined by VN, but there were no significant differences among the three groups determined with MEA. At 3 months, there was a trend towards increasing OPR with ticagrelor (0.0% to 12.5%, *p* = 0.021) as well as prasugrel (21.1% to 50.0%, *p* = 0.008), whereas OPR rates with clopidogrel remained higher (57.5% to 65.0%, *p* = 0.491) as determined by VN (Table 3, Figure 2). With MEA, there was a different trend towards ticagrelor from 1 month to 3 months (30.0% to 47.5%, *p* = 0.105) but a significantly higher OPR rate with prasugrel (15.8% to 42.1%, *p* = 0.020). OPR rates for all platelet function measurements are shown in Table 3 and Figure 2. The Sankey diagram depicts the 13 patients (34.2%) in the prasugrel group that changed from low platelet reactivity (LPR) to OPR with VN; with 11 patients (28.9%) in the prasugrel group and 10 patients (25.0%) in the ticagrelor group changing from LPR to OPR with MEA (Figure 2). According to the two platelet function tests, both prasugrel and ticagrelor showed strong platelet function inhibition. There were no HPR patients in the new P2Y12 inhibitor groups at 1 month as well as 3 months, and most patients were showing LPR at 1 month.

### 3.3. Comparison of All Platelet Function Tests

VN values correlated weakly with MEA (r = 0.334, *p* = 0.000). Regardless of the administration of any P2Y12 inhibitors, the correlation between the measurements appears to be determined by the level of platelet reactivity. In terms of HPR determined by VN, non-HPR patients determined by MEA appeared in relatively greater in numbers. VN and LTA showed a moderate correlation (r = 0.582, *p* = 0.000), while MEA and LTA showed a lower correlation (r = 0.357, *p* = 0.000) (Figure 3).

## 4. Discussion

The findings of this pharmacodynamic investigation can be summarized as follows: (1) higher rates of OPR were observed among clopidogrel-treated patients compared to standard-dose prasugrel or ticagrelor using both platelet function assessments at one month; (2) OPR in the reduced-dose prasugrel and ticagrelor groups was not significantly different versus clopidogrel as determined by MEA assay; (3) compared with standard-dose, reduced-dose prasugrel-treated patients’ OPR rate significantly increased with VN and MEA, and ticagrelor-treated patients’ OPR rate significantly increased with VN; (4) prasugrel and ticagrelor had very low platelet function values compared to clopidogrel at one month and three months according to both platelet function assessments (VN and MEA); (5) prasugrel-treated patients’ platelet reactivity was significantly reduced from one month to three months with both platelet function assessments; (6) the MEA assay might have less sensitivity and consistency than the VN assay. Alternatively, the VN test could be overestimating platelet function relative to the MEA or LTA assay, especially in conditions of strong platelet inhibition.

### 4.1. Importance of OPR in Clinical Practice

Aradi et al., reported that patients with HPR have a 2.7-fold higher risk for ST and a 1.5-fold higher risk for mortality. Meanwhile, patients with LPR have a 1.7-fold higher risk of major bleeding complications compared to those with OPR following PCI, and OPR is therefore associated with net clinical benefits [14].

Our findings show that the OPR rate increased significantly in the prasugrel group and ticagrelor group at 3 months as determined by VN, but when assessed by MEA, the OPR rate was only significantly increased in the prasugrel group. The sub-analysis results from the HOST–REDUCE–POYLTECH–ACS trial showed the percentage of OPR was significantly higher in the de-escalation group (= prasugrel 5 mg) compared to the full dose prasugrel group (61.7 vs.31.7%, *p* < 0.001) and the de-escalation groups had a lower risk of net clinical outcomes [17]. Lee et al., reported that the proportion of patients with OPR was the highest in the 5 mg prasugrel group compared to the conventional-dose prasugrel or ticagrelor groups (90.0% vs. 46.2% vs. 12.5%, *p* < 0.001), similar to our findings [18]. The PRAISE–GENE trial demonstrated that half-dose prasugrel achieved significantly lower PRU values in the peri-procedural period, but there were no statistically significant differences in terms of OPR at 30 days to the clopidogrel group [19]. The A-MATCH trial reported that the 5-mg prasugrel group was more likely to meet normal platelet reactivity (NPR) criteria determined by VN with respect to East Asian criteria for type 2 bleeding. The proportion of NPR was significantly increased in the 5-mg compared to the 10-mg group, and there was a lower incidence of adverse events observed in the NPR group [20]. In addition, the OPR rate in the prasugrel 3.75 mg group was higher than that of the ticagrelor 60 mg group. Compared with the clopidogrel group, the low-dose prasugrel group was associated with a lower incidence of MACE and clinically serious bleeding in Japanese ACS patients [21,22]. Kim et al., reported that compared with standard-dose ticagrelor, half-dose ticagrelor reduced serious bleeding events during the early period of dual-antiplatelet therapy in ACS patients with LPR status determined by VN [23].

The pro-drug clopidogrel is characterized by significant response variability and a substantial proportion of patients exhibit high on-treatment platelet reactivity (HPR). Hence, PFT could serve to safeguard DAPT de-escalation by identifying HPR patients on clopidogrel, as those patients may be exposed to a higher risk of thrombotic events due to insufficient P2Y12 inhibition [24]. In the present study, prasugrel caused a dose-dependent decrease in platelet inhibition from 10 to 5 mg as determined by MEA, while both doses of ticagrelor showed no difference between the 45 and 90 mg twice-daily doses. In contrast, VN showed a dose-dependent increase in platelet reactivity for both prasugrel and ticagrelor. Jin et al., reported that the prasugrel (5 mg) group had a significantly lower average PRU value compared with the 75 mg clopidogrel group at 30 days, while the OPR rate was similar between the 5 mg prasugrel and 75 mg clopidogrel groups [25]. The rate of high on-treatment platelet reactivity (PRU > 235) was significantly lower in the 5 mg prasugrel group than in the 75 mg clopidogrel group. This study demonstrated that 5 mg prasugrel is associated with greater platelet inhibition and lower rates of HPR compared with standard-dose clopidogrel in PCI patients < 75 years old and weighing ≥ 60 kg.

### 4.2. Differences between the 3 Platelet Function Tests

Studies have demonstrated a close link between platelet reactivity during treatment assessed by platelet function tests and clinical outcomes after PCI. We compared standard and reduced doses of prasugrel and ticagrelor with clopidogrel in the maintenance phase of PCI in Korean ACS patients. For STEMI patients, new P2Y12 inhibitors are recommended during DAPT treatment after coronary stent implantation and more STEMI patients used new P2Y12 inhibitors in this trial [3]. Although prasugrel and ticagrelor are superior to clopidogrel for reducing ischemic events in ACS patients [6,7], clopidogrel is still widely used in clinical practice due to its reduced bleeding risk [26]. Our results confirm that prasugrel and ticagrelor elicit significantly higher platelet inhibition than clopidogrel at both standard and reduced doses with the VN and MEA assays. Compared to prasugrel, ticagrelor was a more potent inhibitor as determined by VN, a finding consistent with several prior pharmacodynamics studies [27,28,29]. The correlation among three PFT assays from our results showed that LTA and VN were more highly correlated than MEA. Zhang et al. reported a moderate correlation between LTA and VN before or after PCI, while the correlation between MEA and VN measurements was low [30], which was similar to our findings.

The results for the previously published HOPE–TAILOR trial showed ticagrelor had a lower OPR rate and greater platelet inhibition compared to prasugrel as well as conventional-dose clopidogrel at 1 and 3 months as assessed by VN [10]. The present study included differences between the three methods for measuring platelet reactivity. Contrary to the VN results from the HOPE–TAILOR trial, prasugrel had greater platelet inhibition compared to ticagrelor (and clopidogrel) at 1 and 3 months, as well as a higher OPR rate at 1 month, showing the opposite to the MEA data.

In a study by Alexopoulos et al., significantly lower platelet reactivity was found in patients receiving ticagrelor treatment compared to prasugrel [27]. In the study, platelet function was assessed with ticagrelor (90 mg twice daily) or prasugrel standard-dose at 15 and 30 days after a therapeutic switch to ticagrelor or prasugrel using VN in ACS patients with high on-treatment platelet reactivity on clopidogrel. These results mirror our own findings in terms of the data for testing by VN. However, the prasugrel group showed significantly lower platelet reactivity than for ticagrelor as determined by MEA when using the standard-dose, although there was no difference with reduced doses. The results are also consistent with the findings by Schnorbus et al. Further studies are warranted to confirm this discrepancy [31].

The accurate identification of post-MI patients who might gain further benefit from prolonged DAPT with low-dose ticagrelor remains essential, considering the importance of balancing bleeding and ischemic risk in vulnerable patients [32]. In addition, the identification of PEGASUS and COMPASS phenotypes at baseline based on drug eligibility criteria may help in selecting patients at higher risk of ischemic events who could benefit from more intense treatment [33]. Therefore, treatment de-escalation should be determined on a case-by-case basis.

We acknowledge several limitations to our findings. VN was a standardized measurement, while LTA was measured with two different machines, so consistency between the two results cannot be guaranteed. Secondly, blood sampling was performed in the outpatient clinic, and the exact sampling times will have differed, although it was confirmed that each patient had ingested the medicine (so medication-related differences are addressed). Thirdly, although it was a subgroup analysis of the HOPE–TAILOR study, some patients lacked MEA and LTA data, so there were some differences in baseline characteristics after excluding missing values. Additionally, one of the limitations is the absence of data in terms of long-term thrombotic and hemorrhagic complications. Finally, the small number of patients is a critical limitation of this study, and the longitudinal outcome data have been described previously [10].

## 5. Conclusions

OPR values for reduced-dose prasugrel and conventional-dose clopidogrel at three months were similar but higher than for reduced-dose ticagrelor determined by VN. No differences were noted when assessed by MEA. The MEA assay may be less sensitive and consistent than the VN assay. Further studies are warranted to confirm this discrepancy.

## Figures and Tables

**Figure 1 jcm-11-02480-f001:**
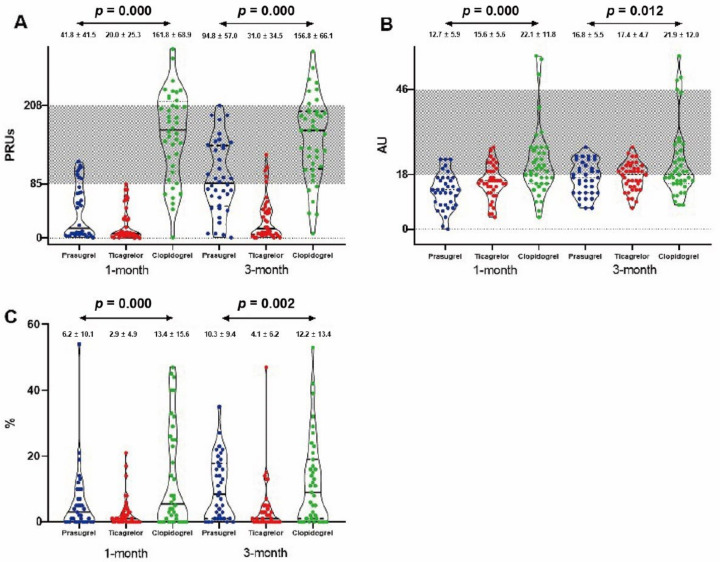
Differences in platelet aggregation inhibition between 1-month and 3-month periods as determined by three different assays ((**A**): VerifyNow; (**B**): MEA; (**C**): LTA). Dotted area represents OPR. PRUs: P2Y12 reaction units; AU: aggregation units; OPR: optimal platelet reactivity.

**Figure 2 jcm-11-02480-f002:**
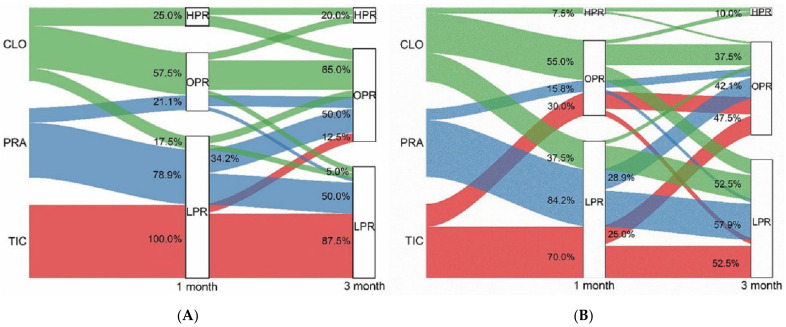
Optimal platelet reactivity rate change from one-month to three-month periods assessed by VN and MEA. Sankey diagram of the relationship between the standard dosage period (1 month) and reduced dosage period (3 months) in platelet function. Each color of the block on the left represents the bar height proportional to the number of patients in each group. The lines connecting the left and right side indicate the relationship between standard dosage and reduced dosage. (**A**): with VerifyNow, (**B**): with MEA. CLO: clopidogrel; PRA: prasugrel; TIC: ticagrelor; HPR: high platelet reactivity; OPR: optimal platelet reactivity; LPR: low platelet reactivity.

**Figure 3 jcm-11-02480-f003:**
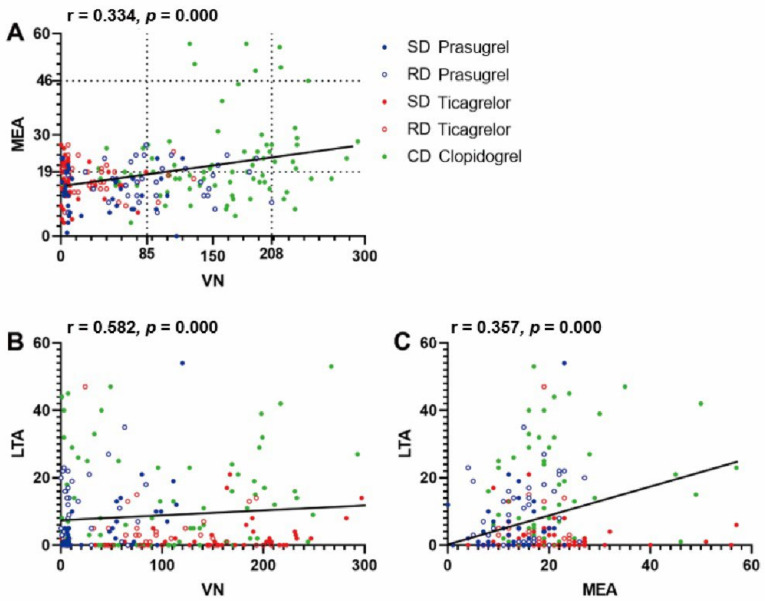
Correlation between three different platelet function tests ((**A**). VN and MEA; (**B**). VN and LTA; (**C**). MEA and LTA). SD: standard dose; RD: reduced dose; CD: conventional dose; VN: VerifyNow; MEA: multiple electrode aggregometry; LTA: light transmittance aggregometry.

**Table 1 jcm-11-02480-t001:** The baseline characteristics of the study patients.

Total Number (*n* = 118)	Prasugrel (*n* = 38)	Ticagrelor (*n* = 40)	Clopidogrel (*n* = 40)
Age, (years)	57.7 ± 10.0	60.8 ± 8.3	63.0 ± 9.9
Gender, (male), *n* (%)	36 (94.7)	34 (85.0)	37 (92.5)
BMI, (kg/m^2^)	24.9 ± 2.5	24.6 ± 2.3	24.7 ± 2.5
LVEF < 50, *n* (%)	14 (36.8)	17 (57.5)	16 (40.0)
Multi-vessel disease, *n* (%)	13 (35.1)	16 (40.0)	17 (42.5)
Risk Factors, *n* (%)			
Hypertension	13 (34.2)	21 (52.5)	18 (45.0)
Diabetes	6 (15.8)	9 (22.5)	14 (35.0)
Dyslipidemia	7 (18.4)	10 (25.0)	5 (12.5)
Current smoker	8 (21.1)	9 (22.5)	9 (22.5)
Medical History, *n* (%)			
Previous MI	4 (10.5)	4 (10.0)	6 (15.0)
Previous PCI	4 (10.5)	8 (20.0)	9 (22.5)
Previous CABG	1 (2.6)	2 (5.0)	0 (0.0)
Previous CVA	0 (0.0)	0 (0.0)	0 (0.0)
Clinical Diagnosis, *n* (%)			
UA	7 (18.4)	3 (7.5)	23 (57.5)
NSTEMI	9 (23.7)	10 (25.0)	11 (27.5)
STEMI	22 (60.5)	27 (67.5)	6 (15.0)
Laboratory Index			
CK-MB	32.1 ± 43.9	28.3 ± 31.3	17.5 ± 14.5
Hemoglobin (g/dL)	14.0 ± 2.0	13.6 ± 1.7	13.4 ± 1.8
Creatinine (g/L)	1.0 ± 0.4	1.0 ± 0.4	1.2 ± 0.9
GFR (ml/min/1.73 m^2^)	84.0 ± 23.6	83.5 ± 22.8	81.3 ± 25.3
Platelet count (10^3^/μL)	218.7 ± 46.9	229.8 ± 66.0	216.9 ± 44.6
Post PCI * Base	208.8 ± 51.6	218.7 ± 31.9	214.6 ± 48.6
Post PCI PRU	57.9 ± 66.5	45.8 ± 77.0	139.7 ± 77.3
** Inhibition (%)	75.8 ± 25.8	79.6 ± 35.2	37.6 ± 26.6
Medication, *n* (%)			
Statin	36 (94.7)	38 (95.0)	37 (92.5)
CCB	9 (23.7)	10 (25.0)	11 (27.5)
β-blocker	29 (76.3)	30 (75.0)	27 (67.5)
ACEI + ARB	9 (23.7)	9 (22.5)	12 (30.0)
Diuretic	3 (9.1)	1 (2.5)	2 (5.1)
Proton-pump inhibitor	14 (36.8)	21 (52.5)	14 (35.0)

Data are presented as number (%). ACEI: angiotensin-converting enzyme inhibitors; ARB: angiotensin receptor blocker; BMI: body mass index; CABG: coronary artery bypass grafting; CCB: calcium channel blocker; CVA: cerebrovascular accident; LVEF: left ventricular ejection fraction; MI: myocardial infarction; PCI: percutaneous coronary intervention; PRU: P2Y12 reaction unit. * Base PRU is the estimated value before obtaining the P2Y12 inhibitor. ** % inhibition = [(Base − Test)/Base] × 100.

**Table 2 jcm-11-02480-t002:** Platelet function values during standard and reduced-dose periods.

		Prasugrel (*n* = 38) Mean (95% CI)	Ticagrelor (*n* = 40) Mean (95% CI)	Clopidogrel (*n* = 40) Mean (95% CI)	Within-Visit Comparisons Overall Effect	*p* vs. C Mean (95% CI) *p*-Value	T vs. *p* Mean (95% CI) *p*-Value	T vs. C Mean (95% CI) *p*-Value
VN	1-month	41.8 ± 41.5 (28.1–55.5)	20.0 ± 25.3 (11.8–28.1)	161.8 ± 68.9 (139.7–183.9)	0.000	120.0 (94.4–145.6) *p* = 0.000	21.9 (6.2–37.5) *p* = 0.007	141.9 (118.5–165.2) *p* = 0.000
	3-month	94.8 ± 57.0 (76.1–113.6)	31.0 ± 34.5 (19.9–42.0)	156.8 ± 66.1 (135.6–177.9)	0.000	61.9 (34.0–89.8) *p* = 0.000	63.9 (42.4–85.4) *p* = 0.000	125.8 (102.2–149.4) *p* = 0.000
* *p*-value		0.000	0.108	0.740				
MEA	1-month	12.7 ± 5.9 (10.7–14.7)	15.6 ± 5.6 (13.7–17.4)	22.1 ± 11.8 (18.3–25.9)	0.000	9.4 (5.2–13.7) *p* = 0.000	−2.9 (−5.6–0.3) *p* = 0.031	6.5 (2.3–10.7) *p* = 0.003
	3-month	16.8 ± 5.5 (14.9–18.7)	17.4 ± 4.7 (15.9–18.9)	21.9 ± 12.0 (18.1–25.7)	0.012	5.1 (0.9–9.3) *p* = 0.018	−0.6 (−2.9–1.8) *p* = 0.638	4.6 (0.5–8.6) *p* = 0.029
* *p*-value		0.003	0.134	0.948				
LTA	1-month	6.2 ± 10.1 (2.7–9.6)	2.9 ± 4.9 (1.3–4.5)	13.4 ± 15.6 (8.2–18.5)	0.000	7.2 (1.1–13.3) *p* = 0.021	3.3 (−0.4–6.9) *p* = 0.080	10.5 (5.1–15.8) *p* = 0.000
	3-month	10.3 ± 9.4 (7.1–13.5)	4.1 ± 8.2 (1.4–6.7)	12.2 ± 13.4 (7.9–16.6)	0.002	1.9 (−3.4–7.3) *p* = 0.470	6.3 (2.2–10.3) *p* = 0.003	8.2 (3.2–13.2) *p* = 0.002
* *p*-value		0.077	0.465	0.732				

* *p*-value compared between 1-month and 3-month periods; *p*: prasugrel; C: clopidogrel; T: ticagrelor; VN: VerifyNow; MEA: multiple electrode aggregometry; LTA: light transmittance aggregometry.

**Table 3 jcm-11-02480-t003:** Platelet reactivity rate during standard- and reduced-dose periods.

			Prasugrel (*n* = 38)	Ticagrelor (*n* = 40)	Clopidogrel (*n* = 40)	Overall Effect	*p* vs. C *p*-Value	T vs. *p* *p*-Value	T vs. C *p*-Value
VN	1-month	HPR	0 (0.0)	0 (0.0)	10 (25.0)				
		OPR	8 (21.1)	0 (0.0)	23 (57.5)	0.000	0.001	0.000	0.002
		LPR	30 (78.9)	40 (100.0)	7 (17.5)				
	3-month	HPR	0 (0.0)	0 (0.0)	8 (20.0)				
		OPR	19 (50.0)	5 (12.5)	26 (65.0)	0.000	0.180	0.000	0.000
		LPR	19 (50.0)	35 (87.5)	6 (15.0)				
* *p*-value			0.008	0.021	0.491				
MEA	1-month	HPR	0 (0.0)	0 (0.0)	3 (7.5)				
		OPR	6 (15.8)	12 (30.0)	22 (55.0)	0.002	0.001	0.030	0.153
		LPR	32 (84.2)	28 (70.0)	15 (37.5)				
	3-month	HPR	0 (0.0)	0 (0.0)	4 (10.0)				
		OPR	16 (42.1)	19 (47.5)	15 (37.5)	0.596	0.711	0.314	0.540
		LPR	22 (57.9)	21 (52.5)	21 (52.5)				
* *p*-value			0.020	0.105	0.116				

* *p*-value compared between 1-month and 3-month periods; *p*: prasugrel; C: clopidogrel; T: ticagrelor; VN: VerifyNow; MEA: multiple electrode aggregometry.

## Data Availability

Data are the property of the authors and can become available by contacting the corresponding author.

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
