# Peer review of "Differences in Optimal Platelet Reactivity after Potent P2Y12 Inhibitor Treatment in Acute Coronary Syndrome Patients Undergoing Percutaneous Coronary Intervention"

_jcm, 2022, doi:10.3390/jcm11092480_

Round 1

Reviewer 1 Report

the authors have not responded adequately to the questions raised

Reviewer 2 Report

I have no further comments. Thank you!

Reviewer 3 Report

The manuscript is improved. I have no further comments. 

This manuscript is a resubmission of an earlier submission. The following is a list of the peer review reports and author responses from that submission.

Round 1

Reviewer 1 Report

The work by Song et al. aim to compare the platelet reactivity of patients who have suffered an acute coronary syndrome and who have undergone percutaneous intervention. Patients were divided in three groups based on the different P2Y12 receptor inhibitors used for their treatment. The authors show the results at two time points, one month and three months post-percutaneous intervention and evaluated platelet reactivity with three different methods (VerifyNow, MEA and LTA). In essence and a priori, the study seems interesting to elucidate which antiplatelet agent could be more useful in the clinic when treating patients, especially of East Asian ethnicity. However, I see important limitations to the study in order to publish in Journal of Clinical Medicine.

Major comments

- The abstract is too long, especially the results section, and it becomes difficult to follow. A good abstract must be concise, attract the attention of the reader, and limit many numerical data that will later be developed in the manuscript. In addition, there are data that do not correspond to the manuscript, such as the study groups or light transmittance aggregometry is not mentioned.

-The introduction does not state a clear hypothesis, only talks about the general objective of the work without mentioning the expected results. Specific objectives are not well established.

- I detect important methodological shortcomings. Authors do not show platelet reactivity baseline values of patients when they were undergone to percutaneous coronary intervention, so it is impossible to know whether the three groups start from the same physiological conditions as they have been randomised. This is a major methodological flaw.

- I am very surprised because the authors use only double antiplatelet therapy in one of the groups, or at least is what they describe in the population methodological section (being different from the abstract). However, in figures and tables they do not refer to the treatment with aspirin.

- In Table 1, it is not described how long it has been since patients have suffered an ACS.

- According to the authors, all the subjects included in the study had suffered an ACS, however, in Table 1 they describe that only four had suffered a previous myocardial infarction. The data does not add up if you look at the UA, NSTEMI and STEMI data later in the table.

Minor comments

-In the manuscript´s title, the authors define prasugrel and ticagrelor as new inhibitors of the P2Y12 receptor when they are drugs with more than a decade already approved and marketed.

- Diabetic individual is a very special population since have high platelet hyperagregability, as a suggestion, a subanalysis of the results could be made without these patients, who make up a small percentage of the population, to avoid possible biases in the study.

Reviewer 2 Report

The authors presented data about differences in OPR after new P2Y12i treatment in ACS patients undergoing PCI. The topic is of interest. The authors found that OPR values for reduced-dose prasugrel and conventional-dose clopidogrel at three months were similar but higher than for reduced-dose ticagrelor determined by Verify Now, but no differences were noted when assessed by multiple electrode aggregometry.

General comments:

1) While the paper is within the word limit, I feel that readability would be enhanced by being more succinct and shortening the word count.

2) The manuscript presentation should be improved. Please, correct typos and grammatical errors in the manuscript.

3) A major revision by a native English speaker is required.

Detailed comments:

4) Please describe in more detail the inclusion and exclusion criteria.

5) Please in the introduction specify that DAPT is used in acute coronary syndromes, even in patients not undergoing PCI (Neumann FJ; ESC Scientific Document Group. 2018 ESC/EACTS Guidelines on myocardial revascularization. Eur Heart J. 2019 Jan 7;40(2):87-165. doi: 10.1093/eurheartj/ehy394.)

6) During the 3-month follow-up, is there any definite data regarding thrombotic or hemorrhagic complications? Please give more information on this point.

7) Is there any data regarding the complexity of revascularization (stents implanted, lesions treated, left main, bifurcation stenting)?

8) Patients enrolled included some with prior MI, PCI, or CABG. Is there any information regarding antiplatelet therapy of these subjects?

9) In addition to the discrepancy between the various methods of assessing platelet reactivity, one of the limitations is the absence of data in terms of long-term thrombotic and hemorrhagic complications.

10) The authors use a ticagrelor dosage of 45 mg twice daily. On what basis did they choose this dosage?

11) Currently, the only approved reduced dose appears to be ticagrelor 60 mg twice daily based on the results of the PEGASUS study. There are real-life experiences in the literature showing its safety in terms of bleeding. Please briefly discuss this point by citing this reference: J Cardiovasc Pharmacol. 2020 Aug;76(2):173-180. doi: 10.1097/FJC.0000000000000856; and eligibility for this treatment in patients with ACS: Int J Cardiol 2021 Dec 15;345:7-13. doi: 10.1016/j.ijcard.2021.10.138.

12) How do the authors think the study results will affect clinical practice?

13) Please specify the acronyms since their first appearance in the main text and abstract.

Reviewer 3 Report

An excellent effort!

Author Response

Thanks for your review.

Round 2

Reviewer 2 Report

The manuscript is improved. I have no further comments.